# Polymeric Nanoparticles Enable mRNA Transfection and Its Translation in Intervertebral Disc and Human Joint Cells, Except for M1 Macrophages

**DOI:** 10.3390/pharmaceutics16040438

**Published:** 2024-03-22

**Authors:** Katrin Agnes Muenzebrock, Fiona Y. W. Ho, Adriano P. Pontes, Carla Jorquera-Cordero, Lizette Utomo, Joao Pedro Garcia, Paul C. Willems, Tim J. M. Welting, Jaap Rip, Laura B. Creemers

**Affiliations:** 1Department of Orthopedics, University Medical Center Utrecht, 3584 CX Utrecht, The Netherlands; k.a.munzebrock@umcutrecht.nl (K.A.M.);; 220Med Therapeutics BV, 2333 BD Leiden, The Netherlands; pontes@20medtx.com (A.P.P.); rip@20medtx.com (J.R.); 3Department of Chemistry and Pharmaceutical Sciences, Vrije Universiteit Amsterdam, 1081 HV Amsterdam, The Netherlands; 4Department of Clinical Sciences, Faculty of Veterinary Medicine, Utrecht University, 3584 CS Utrecht, The Netherlands; 5Department of Orthopedic Surgery, Maastricht University Medical Center, 6229 HX Maastricht, The Netherlands; p.willems@mumc.nl (P.C.W.); t.welting@maastrichtuniversity.nl (T.J.M.W.)

**Keywords:** osteoarthritis, chronic lower back pain, nanomedicine, mRNA delivery

## Abstract

Chronic lower back pain caused by intervertebral disc degeneration and osteoarthritis (OA) are highly prevalent chronic diseases. Although pain management and surgery can alleviate symptoms, no disease-modifying treatments are available. mRNA delivery could halt inflammation and degeneration and induce regeneration by overexpressing anti-inflammatory cytokines or growth factors involved in cartilage regeneration. Here, we investigated poly(amidoamine)-based polymeric nanoparticles to deliver mRNA to human joint and intervertebral disc cells. Human OA chondrocytes, human nucleus pulposus (NP) cells, human annulus fibrosus (AF) cells, fibroblast-like synoviocytes (FLS) and M1-like macrophages were cultured and transfected with uncoated or PGA-PEG-coated nanoparticles loaded with EGFP-encoding mRNA. Cell viability and transfection efficiency were analyzed for all cell types. Nanoparticle internalization was investigated in FLS and M1-like macrophages. No significant decrease in cell viability was observed in most conditions. Only macrophages showed a dose-dependent reduction of viability. Transfection with either nanoparticle version resulted in EGFP expression in NP cells, AF cells, OA chondrocytes and FLS. Macrophages showed internalization of nanoparticles by particle–cell co-localization, but no detectable expression of EGFP. Taken together, our data show that poly (amidoamine)-based nanoparticles can be used for mRNA delivery into cells of the human joint and intervertebral disc, indicating its potential future use as an mRNA delivery system in OA and IVDD, except for macrophages.

## 1. Introduction

Degenerative joint pathologies such as intervertebral disc (IVD) degeneration leading to chronic low back pain (CLBP) and osteoarthritis (OA) are highly prevalent. They are a major cause of pain and disability in adults globally [1,2,3].

The first-line treatments for both CLBP and OA consist of anti-inflammatory and analgesic drugs and/or physiotherapy. These treatments reduce or alleviate the symptoms to a limited extent only [4,5,6]. Moreover, they fail to inhibit degeneration nor promote regeneration of the affected cartilaginous tissues. In OA, the final treatment, therefore, is joint replacement surgery. Even though good surgical procedures are available, the limited durability of knee implants is an issue, especially for younger patients [7]. In CLBP, partial disc resection or spinal fusion are surgical treatment options [8,9,10]. Yet, especially for spinal fusion, there is a risk of developing pathologies affecting the surrounding vertebra and discs [11].

Once cartilaginous tissue is injured, its self-repair is very limited, primarily because of its avascular nature [12]. Inducing regeneration in cartilage and IVD is therefore a challenging task. Recombinant proteins, in particular growth factors, have been investigated to possibly treat CLBP and OA [13,14]. However, the development of an antigenic immune response during growth factor treatment can reduce its clinical efficacy [15]. Moreover, the production of recombinant proteins can be cost intensive and versatility (e.g., regarding post-translational modification) is limited by the cells in which they are produced [16]. Therefore, gene therapy has come into focus. Initially, viral vectors carrying the corrective DNA or RNA were investigated. However, viral vectors such as oncoretroviruses and lentiviruses integrate into the host cell genome, which raised reasonable safety concerns, as integration is hard to control [17,18]. Viral vectors like adeno and adeno-associated viruses (AAV) do not permanently integrate their genome but also have limitations [17]. The capsid of adenoviruses can mediate a strong inflammatory response and AAV has a very limited packaging capacity of below 5 kilobases [17]. Non-viral gene therapy approaches might therefore be an alternative to efficiently and safely deliver nucleic acids to modulate the cell phenotype and provide stimuli for regenerative cell responses.

To avoid genomic integration, delivery of mRNA was recently proposed [19]. To effectively deliver mRNA into the target cells, vectors or carriers are required. A promising approach is using synthetic nanoparticles. Polymeric nanoparticles are very versatile and can be produced in bulk [20]. To be suitable for in vivo use, they should exhibit minimal cytotoxicity, possess a certain stability in the extracellular environment and allow for sufficient cell uptake, intracellular cargo release and protein expression of the mRNA inside the cell. Poly (amidoamine) nanoparticles (PAA nanoparticles) with repetitive disulfide bridges in their backbone have shown these advantageous features in various cell lines already [21,22]. They show high extracellular stability, are taken up by endocytosis and are able to escape the endosome. Additionally, they exhibit bioresponsive features in the cytosol, where the reducing glutathione molecules break the disulfide bridges of the nanoparticle backbone, resulting in the degradation and delivery of the cargo [22]. PAA-based nanoparticles can transfect primary cells from rat, bovine and human origin [23]. However, no insight into the delivery of mRNA by the nanoparticles to human primary cells of the joint and disc is available. Nanoparticles can additionally be modified by conjugating functional moieties to their polymeric backbone. For instance, the addition of polyethylene glycol (PEG) to the particle surface neutralizes the otherwise positive nanoparticle charge, which decreases the uptake by macrophages and increases colloidal stability [24]. However, a positive charge could also be beneficial for uptake into the negatively charged cartilage or disc matrix [25]. We therefore tested cationic nanoparticles with a positive surface charge and nanoparticles with a non-covalent coating with PGA-PEG polymers resulting in a neutral surface charge. More specifically, we investigated for the first time the potential of PAA-based nanoparticles [26] to deliver mRNA to human primary cells most relevant for OA and CLBP and provide a means to express therapeutical molecules by these cells. Cell types included human nucleus pulposus and annulus fibrosus cells from the IVD as well as human OA chondrocytes and fibroblast-like synoviocytes. Additionally, we have tested M1-like macrophages, as in the synovial lining they are involved in OA progression [27].

## 2. Materials and Methods

### 2.1. Cell Isolation and Cell Culture

The anonymous use of redundant tissue for research purposes is part of the standard treatment agreement with patients in the Diakonessenhuis Utrecht and the University Medical Center Utrecht and was in the UMCU carried out under protocol No. 15-092 of the UMCU’s Review Board of the BioBank for the joint tissues. IVD tissue was obtained as part of the standard postmortem procedure and approved by the medical ethics committee of the UMCU (METC No. 12-364). All material was used in line with the code on the ‘Proper Secondary Use of Human Tissue’, installed by the Federation of Biomedical Scientific Societies. Monocytes were isolated from buffy coats purchased from the national blood bank (Sanquin Blood Bank, Amsterdam, The Netherlands).

#### 2.1.1. Human Nucleus Pulposus and Annulus Fibrosus Cells

Intervertebral discs from 3 patients per tissue (nucleus pulposus: 1 female, 2 male, age: 53 ± 11; annulus fibrosus: 2 male, 1 female, age: 49 ± 3) were dissected in a sterile manner. Pieces of nucleus pulposus and annulus fibrosus tissue were collected separately and minced into pieces of 1–2 mm size. Tissue fragments were washed twice in PBS containing 200 U/mL penicillin/streptomycin (Gibco, Billings, MT, USA), 2.5 μg/mL amphotericin B (Gibco) and 50 μg/mL gentamycin (Gibco). For both tissues, tissue fragments were predigested at 37 °C using 0.2% (*w*/*w*) pronase (Roche Diagnostics, Basel, Switzerland) in Dulbecco’s Modified Eagle Medium (DMEM) (1× + GlutaMAX^TM^-I (+4.5 g/L D-glucose, + pyruvate) (Gibco) plus 200 U/mL penicillin/streptomycin, 2.5 μg/mL amphotericin B and 50 μg/mL Gentamycin) for 1 h and digested at 37 °C in collagenase type 2 (Worthington Biochemical, Lakewood, NJ, USA) (0.05% for nucleus pulposus, 0.1% for annulus fibrosus) in DMEM plus 200 U/mL penicillin/streptomycin, 2.5 μg/mL amphotericin B and 50 μg/mL gentamycin overnight. Cell debris was removed using a 70 μm cell strainer (Roth, Karlsruhe, Germany). Cells were washed twice in DMEM and cultured in cell culture flasks using DMEM (1×) + GlutaMAX^TM^-I (+4.5 g/L D-glucose, + pyruvate) plus 200 U/mL penicillin/streptomycin, 2.5 μg/mL amphotericin B and 50 μg/mL gentamycin) plus 10% heat-inactivated fetal bovine serum (FBS) (Biowest, Nuaillé, France). Cells were expanded from passage 1 on in expansion medium (DMEM (1×) + GlutaMAX^TM^-I plus 10% FBS, 100 U/mL penicillin/streptomycin, 0.2 mM ascorbic phosphate (Sigma-Aldrich, Saint Louis, MO, USA) (adjusted to 400 mOsmol using NaCl for nucleus pulposus cells)). When cells started to attach, 1 ng/mL of basic fibroblast growth factor (R&D systems, Minneapolis, MN, USA) was added to the culture medium.

#### 2.1.2. Human OA Chondrocytes

Human articular chondrocytes from 3 patients (all female, age: 61.5 ± 2) were isolated from articular cartilage from 3 patients with OA undergoing total knee arthroplasty. Cartilage was removed sterilely from the subchondral bone and minced. Tissue fragments were predigested at 37 °C using 0.2% (*w*/*w*) pronase in DMEM (1×) + GlutaMAX^TM^-I (+4.5 g/L D-glucose, + pyruvate) plus 100 U/mL penicillin/streptomycin for 2 h, and digested at 37 °C and 5% CO_2_ in 0.075% collagenase type 2 in DMEM plus 200 U/mL penicillin/streptomycin overnight. Undigested debris was removed using a 70 μm cell strainer, followed by a PBS wash. Cells were subsequently plated and grown in a humidified incubator at 37 °C with an expansion medium consisting of DMEM supplemented with 10% FBS, 0.2 mM ascorbic-2-phosphate, 100 U/mL penicillin and streptomycin and 1 ng/mL of basic fibroblast growth factor. Medium was renewed every 3 days. Cells were expanded until passage one and either cryopreserved or further expanded and used for experiments in passages 2–4.

#### 2.1.3. Human Fibroblast-like Synoviocytes

Human fibroblast-like synoviocytes (FLS) from 3 patients (2 female and 1 male, age: 69 ± 8) were isolated from synovial capsule tissue from 3 patients with OA undergoing total knee arthroplasty. Synovium was sterilely separated from adipose tissue and minced. Tissue was then digested using 2 mg/mL Collagenase IV (Sigma-Aldrich) and 0.08 mg/mL Dispase II (Gibco) in Hank’s Balanced Salt Solution (Gibco) for 2 h or until completely digested. The digest was then fitted through an 18G needle, neutralized with DMEM and debris was removed using a 100 μm, and, thereafter, a 40 μm cell strainer (Roth) and washed once in DMEM. FLS were expanded in DMEM supplemented with 10% FBS, 100 U/mL penicillin and streptomycin and 0.1 mM ascorbic-2-phosphate. Cells were used at passages 3–4 to ensure a high percentage of fibroblast in the cell culture.

#### 2.1.4. Human Macrophages

Primary human CD14^+^ monocytes were isolated from buffy coats (3 female donors, age: 38 ± 12) by density gradient separation using Ficoll Paque Plus (GE Healthcare Life Sciences, Piscataway, NJ, USA) followed by magnetically activated cell sorting using human anti-CD14 microbeads (Miltenyi Biotec, Bergisch Gladbach, Germany), as described previously [28]. The isolated CD14^+^ monocytes were seeded into a 96-well plate at a density of 500,000 cells/cm^2^ in X-VIVO^TM^ 15 medium (Lonza, Basel, Switzerland) with 10% heat-inactivated FBS and 100 U/mL penicillin and streptomycin. After cell attachment (approx. 1 h after seeding), cells were stimulated for 72 h (renewed after 48 h) with 10 ng/mL recombinant human TNF (PeproTech, Cranbury, NJ, USA) and 10 ng/mL recombinant human IFNγ (PeproTech) for polarization towards an M1-like phenotype of cells.

### 2.2. Nanoparticle Formulation and Characterization

The nanoparticles were formulated as described in Pontes et al. [26]. In short: Two monomers were synthesized for the polymer. Cystamine bis(acrylamide) was synthesized as reported by Lin et al. [29]. The second monomer, N1-(7-chloroquinolin-4-yl)-hexane-1,6-diamine (Q6), was produced analogously to the synthesis described by Natarajan et al. [30]. Further, a mixture of 6 and 4-aminobutanol (ABOL) was used to make a random PAAQ co-polymer with the CBA via an aza-Michael reaction in a polar protic solvent system suitable for this type of reaction as reported by Pontes et al. [26]. mRNA-loaded nanoparticles were prepared by mixing poly(amidoamine)-based polymers (ps-PAAQ, 20Medtx) with EGFP mRNA (Cleancap^®^, TriLink Biotechnologies, San Diego, CA, USA or CureVac, Tübingen, Germany) in 10% trehalose 10 mM histidine buffer (pH 6.5) and incubated for at least 15 min before storage in the freezer. ps-PAAQ polymers were prepared at 1.5 mg/mL with an mRNA concentration of 0.06 mg/mL resulting in a loading ratio of 25:1 *w*/*w*. For polyglutamic acid–polyethylene glycol (PGA-PEG)-coated nanoparticles, the coating material (mPEG_5k_-b-PLE_50_, Alamanda Polymers, Huntsville, AL, USA) was added to the mRNA solution in the first step, which was then added to the ps-PAAQ polymers in the same mixing step, using a 1:1 PGA-PEG:ps-PAAQ [*w*/*w*] ratio [26]. Nanoparticles used for internalization experiments were co-loaded with EGFP mRNA and silencer Cy3-labeled negative control siRNA (Invitrogen, Carlsbad, CA, USA) at a 10:1 *w*/*w* ratio.

The resulting nanoparticle size and zeta potential were measured using Multi-angle Dynamic Light Scattering (MADLS) in the Zetasizer Ultra (Malvern Panalytical, Malvern, UK), with three different angles: 175°, 90° and 13°. Samples were diluted ten fold in 10 mM of Histidine 10% Trehalose buffer (pH 6.5) and loaded in a low-volume quartz cuvette (ZEN2112, Malvern). Results were analyzed in the ZS Explorer software (version 3.0, Malvern).

### 2.3. Viability, Cell Number and Transfection Efficiency

Human nucleus pulposus cells and annulus fibrosus cells (P1-3) were seeded into a 96-well plate (Greiner Bio one, Kremsmünster, Austria) at a density of 15,000 cells/cm^2^ using the according expansion medium. Human OA chondrocytes (P2-4) were seeded into a 96-well plate at a density of 20,000 cells/cm^2^ using chondrogenic medium (DMEM (1×) + GlutaMAX^TM^-I (+4.5 g/L D-glucose, +Pyruvate) supplemented with 0.2 mM ascorbic-2-phosphate, 1× ITS-X (10 mg/L insulin, 5.5 mg/L transferrin, 6.7 µg/L sodium selenite, 2 mg/L ethanolamine) (Thermo Fisher Scientific, Waltham, MA, USA), 4.0 g/L human serum albumin (Sanquin, Amsterdam, The Netherlands) and 100 U/mL penicillin/streptomycin). Human fibroblast-like synoviocytes (P2-4) were seeded into a 96-well plate at a density of 15,000 cells/cm^2^ using a fibroblast culture medium. Nucleus pulposus cells, annulus fibrosus cells and chondrocytes were transfected 1 day after seeding for 24 h with EGFP mRNA-loaded nanoparticles, uncoated and PGA-PEG-coated, respectively, in chondrogenic medium with 2 mM HEPES (Gibco). Fibroblast-like synoviocytes were transfected 1 day after seeding for 24 h using the same nanoparticles and mRNA in DMEM (1×) + GlutaMAX^TM^-I (+4.5 g/L D-glucose, +Pyruvate) supplemented with 0.1 mM ascorbic-2-phosphate, 4.0 g/L human serum albumin, 100 U/mL penicillin/streptomycin with 2 mM HEPES. M1-like macrophages were transfected using the same nanoparticles and mRNA in X-vivo and 100 U/mL penicillin/streptomycin and 2 mM HEPES for 24 h after polarization.

Particle concentrations ranged from 20 to 80 μg/mL (corresponding to EGFP mRNA concentrations of 0.8 to 3.2 μg/mL). As a positive control, cells were transfected with EGFP mRNA (3.2 μg/mL) using Lipofectamine MessengerMax^TM^ reagent (Thermo Fischer Scientific) according to the manufacturer’s protocol. Cells cultured in cell-type-specific media with 2 mM were used as the negative control.

In order to analyze transfection efficiency and cell viability, cells were first incubated with 5.0 μM Hoechst 33342 (Thermo Fisher Scientific) for 10 min, washed once with DMEM (1×) + GlutaMAX^TM^-I (+4.5 g/L D-glucose, +Pyruvate) and incubated for 15 min with 0.25 μM SYTOX^TM^ Orange (Thermo Fisher Scientific). After that, cells were imaged with a Leica SP8X confocal microscope and a 20× objective. Five random areas per well were imaged. Total cell number, cells positive for SYTOX^TM^ Orange and cells expressing EGFP were counted using ImageJ’s analyze particle function [31]. The ratio of SYTOX^TM^ Orange-positive cells to Hoechst-positive cells was used to calculate cell viability. Transfection efficiency was calculated by determining the ratio of EGFP-positive to Hoechst-positive cells. Normalized cell number was calculated by dividing by the average cell number of the control. To test the effect of nanoparticle buffer on the cell count of macrophages, CD14^+^ monocytes were isolated and polarized to M1-like macrophages as described above and then stimulated with nanoparticle buffer (10% trehalose 10 mM histidine buffer (pH 6.5)) + 20 nM Hepes for 24 h. Buffer concentrations that are equivalent to the buffer concentration range used in the nanoparticle experiments were investigated. Afterward, the DNA content of the cell lysate was determined using Quant-iT™ PicoGreen™ dsDNA Assay Kit (Thermo Fisher, Waltham, MA, USA) according to the manufacturer’s protocol.

### 2.4. Nanoparticle Internalization

Cell internalization of the nanoparticles was investigated in Fibroblast-like synoviocytes and M1-like macrophages, respectively. Cells of 2 donors each were seeded with the above-mentioned media and cell densities in a 35 mm glass-bottom micro dish (Greiner bio one). Cells were transfected with 60 µg/mL uncoated or PGA-PEG-coated nanoparticles loaded with EGFP mRNA and silencer Cy3-labeled negative control siRNA for 6 and 16 h and stained for Hoechst as described above. Cells were imaged with a Leica SP8X confocal microscope and a 63× objective. In order to quantify nanoparticle internalization, the Cy3 signal that was co-localized with the cell body was measured as the average intensity of the Cy3 signal per cell corrected by the background signal using ImageJ. For fibroblast-like cells, up to 30 cells per donor per group were analyzed. For M1-like macrophages, up to 60 cells per donor per group were analyzed. To display cell outlines, M1-like macrophages were labeled with Vybrant™ DiD Cell-Labeling Solution (Thermo Fisher Scientific) according to the manufacturer’s protocol (20 min incubation of cells with DiD dye).

### 2.5. Statistics

Statistical analysis was performed using IBM^®^ SPSS^®^ Statistics.version 29.0.01. For transfection studies, 3 donors with 3 replicates per condition (*n* = 9) were used. For internalization studies, 2 donors with 2 replicates per donor (*n* = 4) were used. Residuals of outcome values were normally distributed according to QQ plots. A randomized block design including donor and particle concentration was used to test for statistical significance by comparing viability, cell number and transfection efficiency. For viability and cell count Dunnet’s post hoc testing was carried out. For transfection efficiency, a Tukey post hoc test was carried out. To test for statistically significant differences in viability and transfection efficiency between samples treated with nanoparticle and Lipofectamine MessengerMax^TM^ of the same mRNA concentration and to compare means values of control and samples treated with Lipofectamine MessengerMax^TM^ a paired *t*-test was used. To compare the viability and efficiencies of treatment with uncoated and PGA-PEG-coated nanoparticles, a linear mixed model corrected for donor effects including a likelihood ratio test with nanoparticle concentration as fixed and donor as random factor to calculate the *p*-value was used. *p* < 0.05 was considered statistically significant.

## 3. Results

### 3.1. Nanoparticle Characterization

#### Uncoated and PGA-PEG-Coated Nanoparticles Show Different Physical Characteristics

Polymeric nanoparticles were co-formulated with either EGFP mRNA only or a mixture of EGFP mRNA and Cy3-labeled nucleotide to yield nanoparticles with a loading ratio of 25. MADLS measurement showed nanoparticle formation with average sizes, as indicated in Table 1. Uncoated nanoparticles measured 110 nm and multiple resolvable peaks (Appendix A). PGA-PEG nanoparticles measured sizes of around 60 nm and monodisperse distribution. Furthermore, uncoated nanoparticles showed a strong positive zeta potential of at least +20 mV, whereas PGA-PEG nanoparticles showed a near-neutral zeta potential.

### 3.2. Viability

#### 3.2.1. Toxicity of Nanoparticles Is Minor in Chondrogenic Cells and FLS

In order to investigate the influence of nanoparticle transfection on viability, apoptosis and proliferation, viability staining and additional cell counting were performed.

In NP cells, transfection with uncoated nanoparticles showed a significant decrease (*p* < 0.001) in viability (72 ± 17%) for 60 µg/mL of nanoparticle compared to control (96 ± 6%) (Figure 1A). For all other concentrations, viability ranged from 94 ± 7% to 84 ± 16% and showed no significant decrease compared to control. Transfection with PGA-PEG-coated nanoparticles for any of the concentrations (viability range 99 ± 1%–94 ± 7%) showed no significant decrease in viability when compared to control (96 ± 6%). In order to compare the nanoparticles’ performance with already commercially available transfection agents, Lipofectamine MessengerMax^TM^ (abbreviated as Lipofectamine) was used as a positive control. Transfection with Lipofectamine (3.2 µg/mL mRNA) showed a significantly lower viability (53 ± 19%, *p* < 0.001) and a significantly lower cell count of 0.5 ± 0.2 (*p* < 0.001, Appendix A) compared to control and compared to the equivalent condition of both nanoparticle types (80 µg/mL nanoparticle = 3.2 µg/mL mRNA). Overall, a significant difference in viability between NP cells treated with uncoated and PGA-PEG-coated nanoparticles was observed (*p* = 0.022). No viability difference between nanoparticle types was observed in any other tested cell type.

In annulus fibrosus (AF) cells, the viability of control cells was 83 ± 12% (Figure 1B). In samples transfected with uncoated nanoparticles, viability ranged between 81 ± 14% and 74 ± 17% for nanoparticle concentrations between 20 and 80 µg/mL and showed no significant difference compared to control. For PGA-PEG-coated nanoparticles, a statistically significant decrease (*p* = 0.013) of viability to 67 ± 8% for the highest concentration was found. Lipofectamine transfection (3.2 µg/mL mRNA) showed significantly lower viability (54 ± 18%) when compared to control (*p* = 0.009) and to transfection with either nanoparticle type (80 µg/mL) (*p* = 0.009 for both). No effects on cell count were seen (Appendix A).

In the control, OA chondrocytes, viability after 24 h was 89 ± 8% and did not significantly change for chondrocytes transfected with uncoated or PGA-PEG-coated nanoparticles of any of the tested concentrations (Figure 1C). In PGA-PEG-coated nanoparticle transfected cell culture at 80 µg/mL nanoparticle, a significantly higher cell count (2.3 ± 1.8, *p* = 0.001) compared to control was seen (Appendix A). As for NP cells, viability for Lipofectamine-transfected OA chondrocytes appeared lower (71 ± 12%), but the decrease was not statistically significant when compared to control or cells transfected with the nanoparticles (80 µg/mL) of either type. Lipofectamine reduced cell count significantly (0.3 ± 0.2, *p* = 0.02) compared to control.

In FLS, the viability of control was 99 ± 1% (Figure 1D). Transfection with uncoated nanoparticles had no significant effect on viability. Transfection with PGA-PEG-coated nanoparticles showed significantly lower viability (87 ± 15%, *p* = 0.002) and cell count (0.7 ± 0.3, *p* = 0.02) for the highest concentration only. Transfection with Lipofectamine showed significantly lower viability (81 ± 20%, *p* = 0.032) and cell count (0.2 ± 0.1, *p* < 0.001, Appendix A) compared to control.

#### 3.2.2. Toxicity in Macrophages Is Dose-Dependent

In M1 macrophages, viability seemed to decrease in a concentration-dependent manner for both nanoparticle types. Transfection of M1-like macrophages with uncoated nanoparticles showed a significantly decreased viability for 60 (79 ± 21%, *p* = 0.044) and 80 µg/mL (67 ± 25%, *p* < 0.001) compared to control (98 ± 1%) (Figure 1E). Transfection with PGA-PEG-coated nanoparticles resulted in significantly decreased viability in the highest concentration (63 ± 30%, *p* < 0.001) compared to control. Viability in the Lipofectamine condition (90 ± 3%) was significantly higher compared to the equivalent mRNA concentration of either nanoparticle type (*p* ≤ 0.04 for both). Transfection with Lipofectamine showed no significant decrease in viability compared to control. Transfection with either nanoparticle type increased cell count significantly to 2.3 to 3.1 for the 3 lower concentrations (*p* (uncoated) < 0.05 and *p* (PGA-PEG-coated) ≤ 0.002). Lipofectamine also significantly increased macrophage cell count to 1.7 ± 0.8 (*p* = 0.01, Appendix A). To exclude that the nanoparticle buffer induced this cell count discrepancy, M1-like macrophages from one donor were treated with nanoparticle buffer only. DNA quantification, however, showed no differences between cells treated with any nanoparticle buffer concentration used compared to control (Appendix A).

**Figure 1 pharmaceutics-16-00438-f001:**
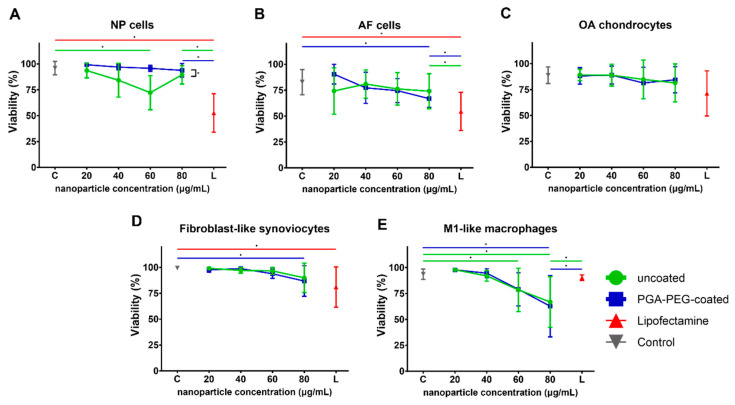
Viability of cells as total Hoechst positive cell count (100%) minus percentage of SYTOX^TM^ Orange positive cells for (**A**) NP cells, (**B**) AF cells, (**C**) OA chondrocytes, (**D**) fibroblast-like synoviocytes and (**E**) M1-like macrophages. mRNA concentration of Lipofectamine control (red) is 3.2 µg/mL. Statistical significance (*p* ≤ 0.05) is displayed by * above the line.

### 3.3. Transfection Efficiency

#### 3.3.1. Transfection Shows EGFP Expression in Chondrogenic Cells and FLS

In NP cells, incubation with both nanoparticle types, as well as Lipofectamine, resulted in the production and detection of EGFP and, hence, successful transfection. For uncoated nanoparticles, the lowest concentration reached a transfection efficiency of 22 ± 15.0% (Figure 2A). Transfection increased significantly for 40 (53 ± 11%, *p* = 0.002) and 80 µg/mL nanoparticles (57 ± 16%, *p* < 0.001) compared to the lowest concentration. Transfection with PGA-PEG-coated nanoparticles showed very low transfection efficiency for the lowest concentration (3 ± 3%). Yet, doubling the concentration significantly increased the transfection efficiency (22 ± 12% *p* < 0.001). Further increasing the nanoparticle concentration to 60 or 80 µg/mL led to a significant increase in transfection efficiency to 48–58% (*p* < 0.001 for both). Comparing the overall transfection efficiency of uncoated and PGA-PEG-coated nanoparticles, a significant difference can be seen (*p* < 0.001, Figure 2A). Both nanoparticle types showed significantly better transfection than Lipofectamine (40 ± 7%, *p* < 0.001) when transfected with the same concentration of mRNA.

Uncoated nanoparticles also showed significant transfection in AF cells (Figure 2B). Transfection efficiency reached 25 ± 18% for 20 µg/mL and significantly increased up to about 45% for 60 µg/mL (*p* = 0.015) and 80 µg/mL (*p* = 0.026). Similarly to NP cells, PGA-PEG-coated nanoparticles did show very low transfection for the lowest nanoparticle concentration (1.0% ± 1.2). For 40 µg/mL, transfection efficiency increased significantly to 19 ± 5% (*p* < 0.001), which further significantly increased to almost 40% for 60 µg/mL and 80 µg/mL (*p* < 0.001 for both). Lipofectamine treatment resulted in significantly lower transfection efficiency (24 ± 5%, *p* < 0.001) when compared to the corresponding condition from both nanoparticle types. No significant difference between the transfection efficiencies of uncoated and PGA-PEG-coated nanoparticles overall was observed.

In OA chondrocytes, transfection with uncoated nanoparticles resulted in successful mRNA delivery for all four concentrations ranging from 14–30% transfection efficiency (Figure 2C). Treatment with PGA-PEG-coated nanoparticles resulted in a transfection efficiency of 7 ± 8% for the lowest concentration. An increase in nanoparticle concentration to 40 and 60 µg/mL resulted in significantly higher transfection efficiency (31%, *p* < 0.001 for both). Lipofectamine-based transfection resulted in 36 ± 18% transfection, which is significantly higher than the efficiency of the corresponding condition with PGA-PEG-coated nanoparticles (18 ± 9%, *p* = 0.05). A significant difference can be seen between the overall transfection efficiencies of uncoated and PGA-PEG-coated nanoparticles (*p* = 0.044, Figure 2C).

Transfection of FLS with uncoated nanoparticles resulted in a transfection efficiency of 62 ± 6% at 20 µg/mL that significantly increased to 78 ± 7% for 80 µg/mL (*p* = 0.005), which was also significantly higher than Lipofectamine *p* = 0.018 (Figure 2D). Transfection with PGA-PEG-coated nanoparticles showed similar transfection efficiency for 20 µg/mL (65 ± 10%). Yet, at 60 and 80 µg/mL, it significantly decreased to 46–52% (*p* < 0.02). Comparing the transfection efficiency of uncoated and PGA-PEG-coated nanoparticles as a total, a significant difference can be seen between the two particle types (*p* < 0.001, Figure 2D), indicating a better performance of uncoated nanoparticles.

#### 3.3.2. Transfection Efficiency in Macrophages Is Negligible

In M1-like macrophages, transfection with either nanoparticle type resulted in very low transfection efficiencies of about 1% for all concentrations (Figure 2E). Lipofectamine transfection resulted in 25 ± 6% translation, which was significantly higher than the corresponding nanoparticle conditions (*p* < 0.001).

**Figure 2 pharmaceutics-16-00438-f002:**
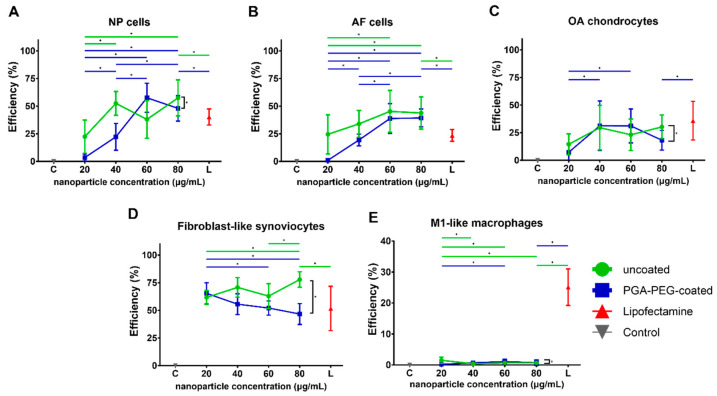
Transfection efficiency of cells analyzed as percentage of EGFP positive cells to total Hoechst positive cell count for (**A**) NP cells, (**B**) AF cells, (**C**) OA chondrocytes, (**D**) fibroblast-like synoviocytes and (**E**) M1-like macrophages. mRNA concentration of Lipofectamine control (red) is 3.2 µg/mL. Statistical significance (*p* ≤ 0.05) is displayed by * above the line.

### 3.4. Internalization of Nanoparticles

#### Macrophages Internalize Nanoparticles

Transfection with the nanoparticles did not result in the successful translation of EGFP in macrophages. Internalization of PAA-based nanoparticles was investigated to elucidate whether it might explain the absence of EGFP expression in macrophages. Lipofectamine MessengerMax^TM^ uptake was not investigated as EGFP expression was shown for all cell types in the previous section. Internalization was defined as the co-localization of Cy3-labelled siRNA with the cell body and was investigated to determine limited uptake as a potential underlying cause. FLS, which showed the highest transfection efficiency was investigated as well. Co-localization of Cy3 signal with the cell body, here displayed as mean Cy3 intensity per cell, was seen in both cell types (Figure 3A and Figure 4A), indicating that both FLS and macrophages take up nanoparticles. In FLS, transfection with uncoated nanoparticles showed cy3 mean intensities per cell of 4.6 ± 1.9, which significantly increased to 12.9 ± 60 after 16 h (Figure 3B). For PGA-PEG-coated nanoparticles, the mean intensity after 6 and 16 h seemed higher (7.4 ± 2.7 and 15.3 ± 4.6) compared to uncoated nanoparticles. 

In M1-like macrophages, internalization was also observed. For uncoated nanoparticles, cy3 mean intensities of 3.2 ± 1.9 and 6.5 ± 1.7 were measured (Figure 4B), whereas for PGA-PEG-coated nanoparticles, mean intensities of 2.4 ± 0.7 and 2.5 ± 0.7 were observed, after 6 and 16 h, respectively (Figure 4B). Due to cell-type-specific cell autofluorescence cut-offs in the range of Cy3 emission determined by imaging the non-transfected control of the according cell type in this analysis, a direct comparison between cell types was not performed.

## 4. Discussion

Our study showed for the first time that poly (amidoamine)-based nanoparticles can be used for intracellular delivery of mRNA in cells of the human joint and intervertebral disc. For human nucleus pulposus cells and annulus fibrosus cells, this is the first report of successful mRNA delivery and expression using a non-viral delivery system. Because currently there are no commercially available polymer-based transfection agents that are optimized for mRNA delivery, Lipofectamine MessengerMax^TM^ was used as a control transfection technique. Both uncoated and PGA-PEG-coated nanoparticles induced no or minimal viability impairment for all tested cell types, except for M1-like macrophages, where a decreased viability was encountered at high nanoparticle concentration. In NP cells, AF cells and FLS, viability only decreased at higher nanoparticle concentrations, for OA chondrocytes no viability changes were seen. At the highest nanoparticle concentrations, viability was significantly higher for the nanoparticles compared to Lipofectamine for NP and AF cells. This overall shows that the tested cell types of the joint and intervertebral disc are tolerating both polymeric nanoparticle types well, especially at the lower concentrations tested. PGA-PEG-coating did not enhance transfection efficiency, only an effect on viability was found in NP cells. Transfection with either nanoparticle variant resulted in the successful translation of the internalized EGFP mRNA in NP cells, AF cells, OA chondrocytes and fibroblast-like synoviocytes. In OA chondrocytes, NP cells and AF cells, a transfection efficiency plateau seemed to be reached at higher nanoparticle concentrations. The maximum transfection efficiency varied per cell type and was higher for fibroblast-like synoviocytes. Similar differences between cell types have also been observed for transfection by the nanoparticles in human bone marrow stromal cells and human synovial stem cells [23]. Cell-type-specific differences in endosomal escape efficiency could be a reason as seen for nanodiamonds in different cell lines [32]. In this study, the transfection efficiency of uncoated and PGA-PEG-coated nanoparticles was tested and compared. For NP cells and AF cells, efficiency at low concentrations seemed higher for uncoated nanoparticles compared to PGA-PEG-coated nanoparticles. For FLS, the uncoated nanoparticles performed better for the higher concentrations. The better performance of uncoated nanoparticles seen in some of the cell types might be explained by their positive charge, which reportedly improves attachment to the cell and subsequent endocytosis [20,33]. Our data differ from previous results showing improved translation using PGA-PEG-coated nanoparticles compared to uncoated nanoparticles in the C28/I2 chondrocyte cell line [34], as well as in muscle tissue in a murine in vivo study [26]. Yet, the PGA-PEG-coated nanoparticles also perform reasonably well. Both nanoparticle versions have their unique advantages that might be beneficial in future research: the positive charge of the uncoated nanoparticles might improve penetration into negatively charged cartilage or IVD matrix, whereas the smaller more stable size and inhibited aggregation of the PGA-PEG-coated nanoparticles might be beneficial for general tissue penetration and retention when injected into the joint space [35]. However, to what extent our results can be extrapolated to in vivo transfection remains to be determined. In NP and AF cells, nanoparticle transfection showed significantly better transfection efficiency than Lipofectamine, indicating that the nanoparticles are a promising tool to deliver mRNA in vitro and appear to be superior to commercially available Lipofectamine MessengerMax^TM^, for in vitro application. Moreover, the high cell viability in combination with the successful transfection supports the use of nanoparticles in vivo.

M1-like macrophages did not show any detectable expression of EGFP protein after transfection with either type of nanoparticle, despite clear internalization. Interestingly, the uptake of uncoated nanoparticles appeared only slightly higher, even though PEG coating is commonly assumed to shield particle uptake by macrophages [36]. However, Lipofectamine was able to transfect the macrophages. One reason for the discrepancy compared to the nanoparticles might be different uptake routes. Lipofectamine-mediated uptake reportedly takes depends on micropinocytosis and phagocytosis [37]. Also, membrane fusion was indicated as an uptake mechanism, as blocking endocytic pathways still showed successful siRNA delivery into dendritic cells [38]. The PAA-based nanoparticles, on the contrary, may be taken up by a different pathway like flotillin-1-dependent endocytosis [39]. In macrophages, flotillin is involved in the efficient functioning of phagolysosomes to fend off pathogens like fungi, which might indicate that the nanoparticles and loaded nucleic acids might similarly be degraded by using this path of cell entry [40]. An approach to enhance transfection and translation in macrophages is to use mRNA resistant to degradation using modified nucleotides, e.g., pseudo pyridine or 5-methyl-cytidine [41]. However, the lack of translation may also be an advantage. If for example, FLS would be the target cell type, off-target effects of gene activity modulation in macrophages can be excluded.

The increase in M1-like macrophage cell number by transfection with nanoparticles or Lipofectamine is, as such, difficult to explain. Previous findings correlated exposure to biomaterial with macrophage proliferation [42]. So, the contact with the transfection agents itself might explain the increased cell number. Generally, successful transfection and low cytotoxicity of the connective tissue cells were shown. However, in vivo, the extracellular matrix of the IVD and cartilage might affect nanoparticle internalization by resident cells. In addition, particle clearance from the joint space might also hinder the effectivity in vivo [43]. In vivo, poly (*N*-isopropyl acrylamide) nanoparticles injected in healthy rat joints were shown to penetrate the cartilage and remain in the tissue for days [44]. Moreover, collagen II antibody-conjugated polymeric nanoparticles were successful at delivering siRNA and stopping the progression of post-traumatic OA in a mouse model [45], supporting the feasibility of such an approach. However, several factors have been shown to affect nanoparticle penetration into dense ECM, such as size, charge and polymer composition [46]. An additional complicating factor is the degree to which nanoparticles are prone to protein corona formation, which increases their size but may also affect uptake otherwise [47,48,49]. It is worth noting that delivery into chondrocytes and NP cells may not be required for all gene targets, as overexpression of a gene encoding a particular secreted modulatory protein may also be achieved by joint cells that are more accessible to nanoparticles. Alternatively, gene modulation can also be achieved by using exogenously added cells pretreated with nanoparticle-delivered nucleic acids and producing the therapeutic protein [50].

In conclusion, the current data show for the first time the feasibility and effectivity of using PAA-based nanoparticles as an mRNA delivery system for cells involved in OA and IVDD in vitro. This work is a necessary basis for future investigations of polymeric nanoparticles for mRNA delivery in OA and CLBP. To overcome the limitations of monolayer culture and bridge the gap to clinical use, 3D culture models or ex vivo tissue cultures should be used to further investigate the feasibility of polymeric nanoparticles for nucleotide delivery in OA and CLBP. Future in vivo studies will provide final evidence for their clinical applicability in these diseases. 

## Figures and Tables

**Figure 3 pharmaceutics-16-00438-f003:**
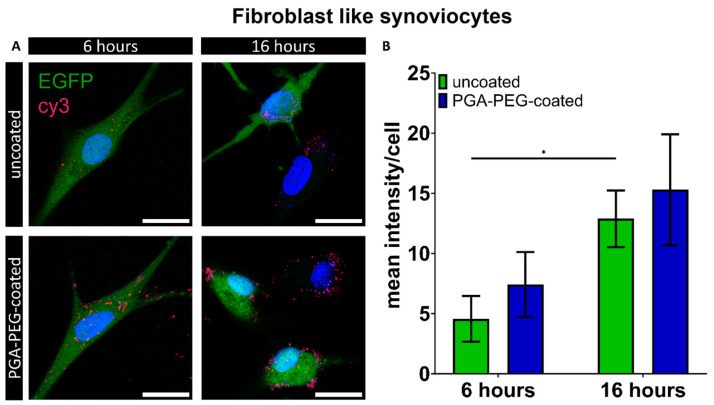
Overview of nanoparticle co-localization of uncoated and PGA-PEG-coated nanoparticles after 6 and 16 h of transfection in fibroblast-like synoviocytes. (**A**) Exemplary maximum projection of Z-stack showing internalization of cy3-siRNA-carrying nanoparticles (red) in FLS cells for uncoated and PEG-coated nanoparticles**. Green: EGFP expression.** Nuclei in blue (Hoechst), scale bar: 20 µm. (**B**) Graphical summary of mean cy3 intensity/cell. Means and SD. Statistical significance (*p* ≤ 0.05) is displayed by * above the line.

**Figure 4 pharmaceutics-16-00438-f004:**
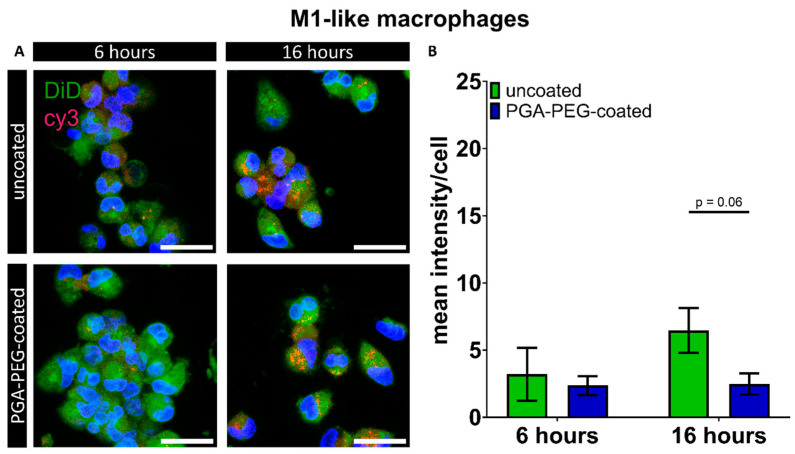
Overview of nanoparticle co-localization of uncoated and PGA-PEG-coated nanoparticles after 6 and 16 h of transfection in M1-like macrophages. (**A**) Exemplary maximum projection of Z-stack showing internalization of cy3-siRNA-carrying nanoparticles (red) in macrophages for uncoated and PEG-coated nanoparticles**. Green: cell staining using DiD** cell labeling was applied to determine colocalization with nanoparticles; as it is in the red spectrum, green pseudo coloring was used to avoid confusion with the Cy 3 signal (yellow emission spectrum, but here depicted in red for better visibility). Nuclei in blue (Hoechst), scale bar: 20 µm. (**B**) Graphical summary of mean cy3 intensity/cell. Means and SD, *p* ≤ 0.05.

**Table 1 pharmaceutics-16-00438-t001:** Particle characterization by Multi-angle Dynamic Light Scattering (MADLS). Results are the combined data of three individual measurements. All nanoparticles were loaded with 60 µg/mL of EGFP mRNA. See Appendix A for particle size distributions.

NP Type	ps-PAAQ:mRNA Ratio (*w*/*w*)	Peak 1 Mean by Intensity (nm)	Polydispersity Index (PDI)	Zeta Potential (mV)
Uncoated	25	112.9	0.341 ± 0.079	+20.4 ± 2.4
Coated	25	58.8	0.105 ± 0.026	−0.1 ± 1.2
Uncoated + Cy3-siRNA	25	108.7	0.317 ± 0.056	+24.8 ± 5.3
Coated + Cy3-siRNA	25	67.4	0.205 ± 0.066	−4.4 ± 1.4

## Data Availability

The datasets generated during and/or analyzed during the current study are available from the corresponding author upon reasonable request.

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
