# Peer review of "Polymeric Nanoparticles Enable mRNA Transfection and Its Translation in Intervertebral Disc and Human Joint Cells, Except for M1 Macrophages"

_pharmaceutics, 2024, doi:10.3390/pharmaceutics16040438_

Round 1

Reviewer 1 Report

Comments and Suggestions for Authors

In this manuscript, the authors investigated the poly (amidoamine)/mRNA nanoparticles transfection efficiency and cytotoxicity on the human joint and intervertebral disc cells such as human OA chondrocytes, human nucleus pulposus (NP) cells, human annulus fibrosus (AF) cells, fibroblast-like synoviocytes (FLS) and M1-like macrophages. They found the PAMAM/mRNA nanoparticles could enter all types of cells and can express GFP protein in all cells except macrophages. This manuscript can be further improved from the following several aspects before being considered to be published. Below are my comments on the manuscript:

Comment 1: In the abstract, please briefly describe why and how “mRNA delivery could halt inflammation and degeneration and induce regeneration”.

Comment 2: On page 3, the first sentence “where the reducing glutathione molecules break the disulfide bridges of the nanoparticle backbone, resulting in degradation and delivery of the cargo” needs more evidence. As far as I know, the PAMAM consists of abundant amide and amine groups. Is there any evidence that the PAMAM has disulfide bridge in the polymer backbone?

Comment 3: Lipofectamine control should be shown in the fluorescent images (figure 3 and figure 4).

Comment 4: according to the authors, macrophages can uptake but not express mRNA containing nanoparticles. Then what is the green fluorescence in figure 4?

Comment 5: transfection efficiency can be analyzed by flowcytometry which is more accurate and unbiased than counting random areas of the fluorescent images. The current method in the manuscript is acceptable. But to strengthen the data reliability, flowcytometry is recommended.

Comment 6: questions to the control selection. The authors used lipofectamine 2000 as the control. There are many other commercial transfection reagents in the market, for example, the lipofectamine 3000 or Lipofectamine MessengerMAX which can achieve more than 90% mRNA transfection efficiency in common cell lines. Why didn’t the authors choose those reagents as controls? Lipo2000’s transfection efficiency can also be optimized since it can achieve more than 80% mRNA transfection efficiency on some common cell lines. In this manuscript, the highest lipo2000’s transfection efficiency is less than 50%. Did the authors optimize the lipo2000/mRNA system according to the manufacture’s protocol?

Another concern about the control selection is: since the manuscript is investigating a polymeric gene vector, why didn’t the authors choose polymer based commercial transfection reagents, for example, jetMESSENGER as the control?

The reason proper controls should be selected is: if some of the commercial mRNA delivery reagents show higher efficiency and lower toxicity, will the proposed PAMAM nanoparticle still be the best candidate for mRNA delivery?

Comment 7: the innovation of this manuscript is not high enough. All polymers, mRNAs, and cells were not new and some of them were well investigated. I agree that this research is useful for some researchers in this field to some extent. But I can hardly believe that the innovation/significance of this manuscript is high enough to attract broad interest in the field of gene delivery.

Comment 8: be careful with some misleading or ambiguous statements. For example, in the abstract, the authors claimed that the nanoparticle “showing promise of nanoparticle use as an mRNA delivery system in OA and IVDD”. However, this manuscript only provided in-vitro experimental data, and OA and IVDD are involved in in-vivo studies. It is not appropriate to raise a statement without real “promising” data. My point is also supported by the authors on page 13: “However, in vivo, the extracellular matrix of the IVD and cartilage might affect nanoparticle internalization by resident cells. In addition, particle clearance from the joint space might also hinder the effectivity in vivo”

Comment 9: only 16 references are up to date (within the past 5 years) out of a total of 45 references. Please include more new references rather than old ones so that this research can be up to date.

Reviewer 2 Report

Comments and Suggestions for Authors

Manuscript is dedicated to study of transfection efficiency of poly(amidoamine)-based polymeric nanoparticles and the particles modified with copolymer of ethyleneoxide. The nature of copolymer is not presented in manuscript- authors use abbreviations PGA and PLE without their explainations.

The manuscript is well organized. However, there are some questions that require additional information.

First, the nature of copolymer should be added to the text.

The procedure of preparation of nanoparticles seems to be shortened. In general, polymeric micelles could not be obtained via simple mixing of the compounds. Otherwise the mixture of polydisperse particles will be obtained. Actually, the size distirbution of non-modified nanoparticles in supplementary matherials reflects this fact. So, the details of the procedure of nanoparticles synthesis should be added, espscially for the narrow distributed microparticles with PEO-based copolymer. The additional confirmation about composition of the obtained nanoparticles should be presented.

Second general question concenrs the purpose of the surface modification of the nanoparticles. In results and discussion sections no significant benefit of utilization of either non-modified particles or PEGylated particles is presented. Please add the corresponding section in discussion.
